# *Ciona Brachyury* proximal and distal enhancers have different FGF dose-response relationships

**Matthew J. Harder, Julie Hix, Wendy M. Reeves, Michael T. Veeman***

Division of Biology, Kansas State University, Manhattan, Kansas, United States of America

* veeman@ksu.edu

## Abstract

Many genes are regulated by two or more enhancers that drive similar expression patterns. Evolutionary theory suggests that these seemingly redundant enhancers must have functionally important differences. In the simple ascidian chordate *Ciona*, the transcription factor Brachyury is induced exclusively in the presumptive notochord downstream of lineage specific regulators and FGF-responsive Ets family transcription factors. Here we exploit the ability to finely titrate FGF signaling activity via the MAPK pathway using the MEK inhibitor U0126 to quantify the dependence of transcription driven by different *Brachyury* reporter constructs on this direct upstream regulator. We find that the more powerful promoter-adjacent proximal enhancer and a weaker distal enhancer have fundamentally different dose-response relationships to MAPK inhibition. The Distal enhancer is more sensitive to MAPK inhibition but shows a less cooperative response, whereas the Proximal enhancer is less sensitive and more cooperative. A longer construct containing both enhancers has a complex dose-response curve that supports the idea that the proximal and distal enhancers are moderately super-additive. We show that the overall expression loss from intermediate doses of U0126 is not only a function of the fraction of cells expressing these reporters, but also involves graded decreases in expression at the single-cell level. Expression of the endogenous gene shows a comparable dose-response relationship to the full length reporter, and we find that different notochord founder cells are differentially sensitive to MAPK inhibition. Together, these results indicate that although the two *Brachyury* enhancers have qualitatively similar expression patterns, they respond to FGF in quantitatively different ways and act together to drive high levels of *Brachyury* expression with a characteristic input/output relationship. This indicates that they are fundamentally not equivalent genetic elements.

## Author summary

When and where genes are expressed is controlled by regulatory DNA regions known as enhancers. Genes often have multiple enhancers that control expression in different cell types or embryonic regions, but there are also genes that have multiple enhancers that

**Data Availability Statement:** All relevant data are within the manuscript and its Supporting Information files.

**Funding:** This work was funded by award 1R01HD085909 from the US National Institutes of

Health (www.nih.gov) to MV. The funders had no role in study design, data collection and analysis, decision to publish, or preparation of the manuscript.

**Competing interests:** The authors have declared that no competing interests exist.

control near-identical expression patterns. These 'shadow' enhancers are common features of many animal genomes, but it is unclear to what extent they are truly identical in function. Here we studied a pair of shadow enhancers for the notochord-specific gene *Brachyury* in the simple model chordate *Ciona* that are both directly activated by the same signaling pathway. We titrated the activity of this pathway with graded doses of a pharmacological inhibitor and measured the effects in quantitative enhancer assays. We found that the two enhancers had significant differences in sensitivity and cooperativity to the same shared regulator and are thus not identical in function when assessed quantitatively. We also identified subtle differences in sensitivity to this upstream signal between different notochord precursor cells.

## Introduction

Embryonic development depends on the precise spatial and temporal regulation of gene expression. Enhancers and other *cis*-regulatory elements embody the logic of the regulatory genome via their specific sets of binding motifs for different sequence-specific transcriptional activators and inhibitors [1,2]. It is now clear that 'shadow,' 'distributed,' or 'redundant' enhancers are ubiquitous features of many genomes [3–5]. These terms refer to cases in which a gene has multiple non-overlapping regulatory elements that control seemingly identical expression patterns. The prevalence of these seemingly redundant elements suggests that they must have important separable functions or else they would not be evolutionarily conserved. Here we refer to these as 'shadow' enhancers, though we note that in current usage this term refers generically to sets of two or more enhancers driving near-identical expression patterns and does not imply a hierarchy of 'main' versus 'shadow' enhancers.

Shadow enhancers are thought to act together to drive high levels of gene expression that can buffer gene regulatory networks against problems resulting from stochastic transcriptional noise, mutation, or environmental perturbation. In support of this, there are several examples in *Drosophila* and vertebrates in which deletions of single shadow enhancers only show a phenotype when grown under heat stress or in sensitized genetic backgrounds [4,6,7]. Predicted shadow enhancers in *Drosophila*, however, show increased sequence conservation compared to solitary enhancers [3], suggesting that they may have separable functions beyond jointly driving higher levels of expression of their regulated gene.

Shadow enhancers in some cases control similar but not completely overlapping expression patterns, with the differences in expression being functionally important. A related idea is that cooperative and/or inhibitory interactions between multiple enhancers can create sharper boundaries of expression than single enhancers [8,9]. Some shadow enhancers have been shown to drive comparable expression patterns using fundamentally different *cis*-regulatory logic involving distinct upstream transcription factors [10], which may provide another aspect of developmental robustness.

In addition to these questions about the overall roles of shadow enhancers, there are also major questions about how regulatory information from multiple enhancers becomes integrated into the expression of the regulated gene. Shadow enhancer pairs have been found to function in sub-additive, additive and super-additive regimes [11], although sub-additive and additive relationships have been most common in the limited number of cases where this has been addressed. Scholes et al [12] recently found that the additivity of different combinations of *Krüppel* enhancers is not uniform as a function of the different concentrations of upstream activators present at different AP positions in the early *Drosophila* embryo. These particular

enhancers are thought to respond to different combinations of upstream transcription factors, but it is also possible that shadow enhancers might have quantitatively different responses to the same upstream regulators.

A mechanistic understanding of how shadow enhancers work together to control gene expression depends on being able to quantify transcriptional input/output relationships in ways that are difficult in most model organisms. Quantitative analyses of shadow enhancer function have generally used the early *Drosophila* embryo as a model, where the AP patterning system provides natural gradients of key transcription factors [8,9,11–13]. In most cases, however, the quantitative details of shadow enhancer function are completely unknown.

The embryo of the ascidian tunicate *Ciona robusta* (formerly *Ciona intestinalis* Type A) provides an excellent alternate model system in which to study transcriptional input/output relationships. Thousands of transgenic embryos can be quickly obtained through simple electroporation methods [14], allowing rapid dissection and analysis of *cis*-regulatory elements in reporter assays. Development proceeds via stereotyped and well-characterized lineages from fertilized egg to a swimming chordate tadpole larva with a muscular tail, dorsal neural tube, and notochord in less than 24 hours [15–17]. Its sequenced genome is only 1/20th the size of the human genome, with around 15000 genes [18].

The T-box transcription factor Brachyury (Bra) is a major regulator of notochord fate in *Ciona* [19–23]. Unlike in vertebrates, it is only expressed in the notochord in ascidians and does not have broader roles in the posterior mesoderm [14,24,25]. *Bra* expression is induced in the presumptive notochord starting at the 64-cell stage through the intersection of lineage-specific transcription factors, including Zic-r.b [26], and the activity of FGF-regulated Ets family transcription factors [21,27]. Two *Bra* enhancers have been identified, including one proximal to the transcription start site [14] and a more distal enhancer several hundred bp upstream [28]. The *Bra* distal enhancer was the first reported shadow enhancer in *Ciona*, but a recent reporter assay survey of open chromatin regions suggests that they are quite common [5]. Reporter constructs for both of these *Bra* enhancers have very similar expression patterns including specific expression in both the primary and secondary notochord and ectopic expression in the mesenchyme [14,28].

FGF is expressed on the vegetal side of the embryo under the control of maternal beta-catenin signaling, and plays a key role in establishing a large number of distinct vegetal and marginal cell fates including notochord as reviewed by [29]. The MEK inhibitor U0126 has been widely used in ascidians to interfere with these inductive interactions [22,27,30–36]. No off-target effects have been described and no other MAPK-dependent ligands have been identified at these stages [21,37]. Simultaneous knockdown of FGF9/16/20 and FGF8/17/18 eliminates *Bra* expression similarly to U0126 treatment, confirming that U0126 effects on notochord fate are specific to the inhibition of the FGF signaling pathway [27].

Previous studies using U0126 in ascidian embryos have always used high doses with the goal of completely blocking MAPK pathway activity. Here we systematically titrate MAPK pathway activity using finely graded doses to test the hypothesis that the *Bra* Proximal and Distal enhancers have quantifiably different input/output relationships. We find that the *Bra* Proximal and Distal enhancers each have a distinct, characteristic response to graded FGF pathway inhibition, and that they appear to act in a weakly super-additive fashion. Using *in situ* hybridization, we show that expression of endogenous *Bra* has largely similar responses to those seen in the reporter assays, but also that the different precursor cells of the notochord at gastrulation have subtle but detectable differences in their response to FGF inhibition.

## Results

### Qualitative responses to U0126

Recent ATACseq data [5] identifies several open chromatin regions upstream of the *Bra* transcription start site at timepoints that overlap the onset of *Bra* expression (Fig 1A). We cloned the three open chromatin regions closest to the transcriptional start site into a standard *Ciona* reporter vector containing a minimal basal promoter from *Ciona Friend of Gata* (bpFOG) and a Venus YFP reporter gene. We found that the peak overlapping the start site and the small peak ~500 bp upstream each drove reasonably strong reporter expression in the notochord and weaker expression in the mesenchyme. These correspond to previously established enhancer regions from [14] and [28] respectively, and we refer to them here as the *Bra* 'Proximal' and 'Distal' enhancers in reference to their positions in the genome. This nomenclature is purely descriptive and is not meant to imply that distance from the *Bra* transcriptional start site plays a direct role in any quantitative differences between them. We also generated a Full Length reporter that included both enhancers and the genome sequence spanning the gap between them (Fig 1B). The third, even more distal ATACseq peak located around the KH2012.S1404: 4800–5100 bp positions did not drive reporter expression in any of our assays, but we cannot exclude it having some form of yet to be determined regulatory activity.

Bra expression requires both the transcription factor Zic-r.b as well as active FGF signaling mediated by Ets family transcription factors [21,26](Fig 1C). These interactions are likely direct because mutation of predicted Zic and Ets binding sites in both the Proximal [38] and Distal [28] enhancers abrogates Bra expression. Essential Zic and Ets sites have also been identified in the *Bra* enhancer in the stolidobranch ascidian *Halocynthia* [39]. There may potentially be other direct upstream inputs into these two enhancers, but Zic and Ets family transcription factors are well-established direct activators of both. To identify quantitative differences between the *Bra* Proximal and Distal enhancers, we systematically titrated the FGF signaling pathway by the addition of varying doses of the MEK inhibitor U0126 at the 16-cell stage to embryos electroporated with the Proximal, Distal, or Full Length reporters. Embryos were fixed at the early-tailbud stage (Hotta stage 19) [17], then stained, cleared and imaged *in toto* by confocal microscopy (Fig 1D). Embryos were stained by antibody for the Venus reporter to provide a bright signal of reporter expression that was not confounded by fluorescent protein maturation times and had minimal photobleaching. Embryos were also stained with phalloidin to visualize embryonic morphology.

Electroporated transgene expression in *Ciona* is mosaic and subject to variable transfection efficiency between different electroporations. We controlled for this in several ways. We only included replicates where the capacitance reported by the electroporator in time constant mode was between 900 and 1300 μFd. The reasons for variable capacitance are unclear, but we have empirically found that electroporations within these bounds tend to have consistent and robust expression. We imaged a sample of ~10 embryos for each construct/dose combination in each of at least three biological replicates. Embryonic morphology becomes severely perturbed at high doses of U0126, but characteristic morphological phenotypes representative of each dose could still be identified based on phalloidin staining. We selected embryos to image at random but excluded embryos with obvious non-specific phenotypes that were not representative of U0126 treatment. The experimenter was completely blind to reporter expression while selecting the embryos to be imaged. We performed at least three independent biological replicates based on separate fertilizations and electroporations for each construct/dose combination, and included a DMSO control treatment for all electroporations.

Very low doses of U0126 have minimal effects on embryo morphology, but defects become more common and pronounced as the dose increases, with severe notochord malformation

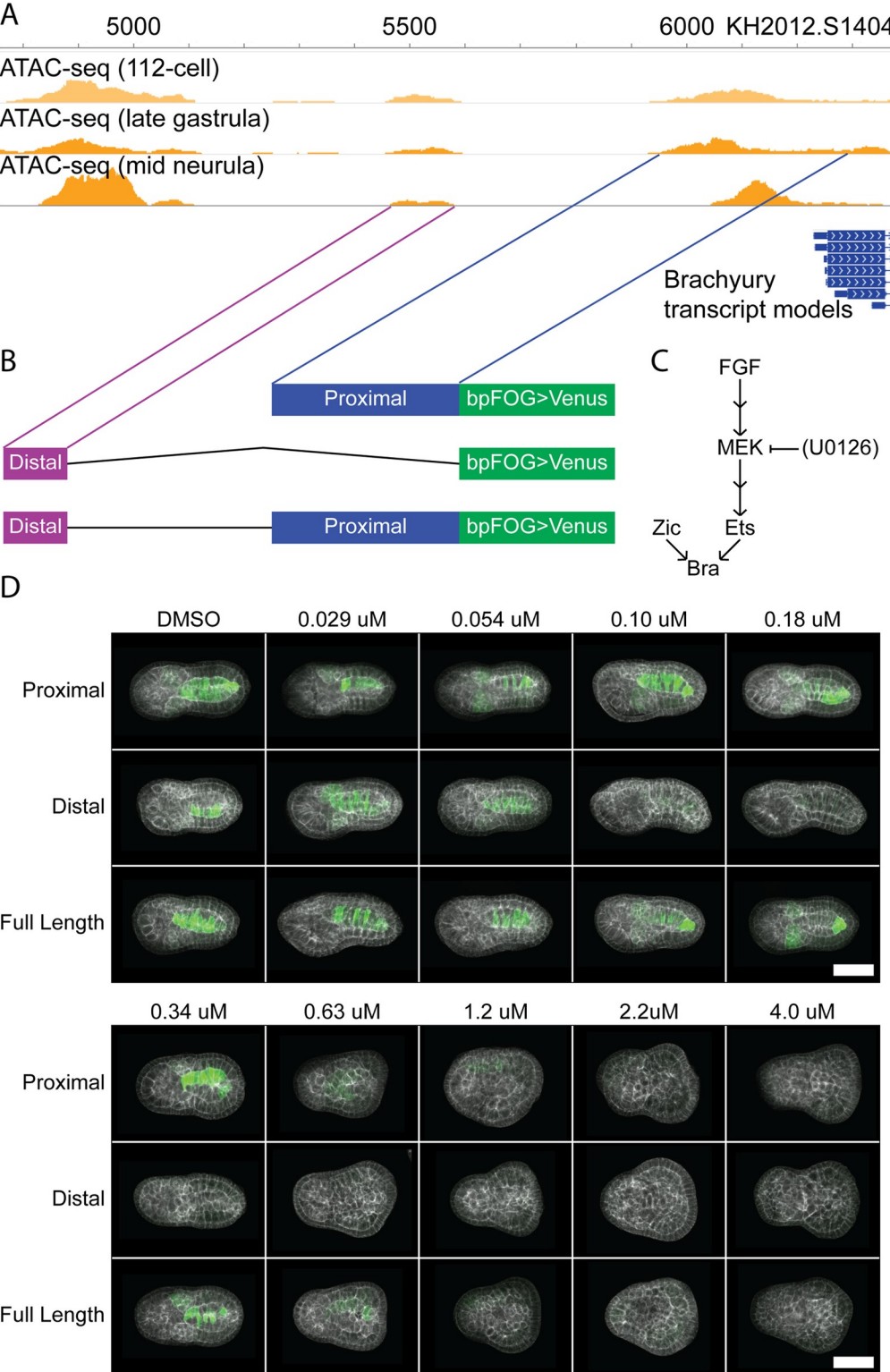

**Fig 1. U0126 inhibition of *Bra* reporter constructs. (A)** ATACseq tracks for the genomic region immediately upstream of *Bra* in wildtype *Ciona* embryos at the 112-cell, late gastrula, and mid neurula stages. **(B)** Schematic of reporter constructs used. A 340-bp Proximal enhancer, 116-bp Distal enhancer, and a construct spanning both enhancers with the wild-type 371-bp spacer region between them, each fused directly to the FOG basal promoter in front of a Venus YFP coding sequence. The lines connecting the Proximal and Distal enhancers to the ATACseq tracks

in (**A**) indicate the genomic location of each enhancer. (**C**) Simplified schematic of the upstream regulators of *Bra* expression. U0126 inhibits FGF signaling at the step of MEK phosphorylating ERK. Double arrows indicate omitted pathway steps. (**D**) Representative images of the three reporter constructs at the indicated U0126 doses. Each image represents a shallow sum of slices through the notochord or the equivalent depth of the embryo at doses where notochord morphology has been lost. Green: reporter. White: phalloidin. Scale bar: 50 microns.

becoming frequent at the 0.34 μM dose, and notochord cells becoming almost impossible to identify morphologically by the 0.63 μM dose (Fig 1D). In embryos electroporated with either the Proximal or Full Length reporters, expression persisted to some extent even as the notochord itself became otherwise unrecognizable. Conversely, the Distal reporter appeared far more sensitive to U0126 treatment, with expression becoming dramatically reduced even at very low doses that did not produce frequent notochord malformation phenotypes. At the highest doses of 1–4 μM, expression of all three reporters was essentially eliminated. While we did not directly quantify any aspect of MAPK pathway activity across this range of doses, our assumption is that MAPK pathway activity is monotonically inhibited by U0126. A quantitative understanding of the U0126 dose/response curve with respect to ERK phosphorylation or other readouts of pathway activity is not needed to identify quantitative differences between the three *Bra* enhancer constructs.

## *Bra* enhancers have distinct dose-response relationships to U0126

To quantify reporter expression within each embryo, we summed total background-subtracted expression over the whole of each imaged embryo and normalized and scaled the data based on vehicle (DMSO) controls to account for variable electroporation efficiency (Fig 2A, details in Methods). We used bootstrap estimates for all statistical comparisons with no underlying assumptions about normality or homoskedasticity. Data files and analysis scripts are in S1 Appendix. We first compared the quantitative expression of the three constructs in the DMSO controls. This confirmed our qualitative assessment that the Proximal enhancer is stronger than the Distal enhancer, and also showed that the Full Length reporter expression is slightly higher than the sum of the Proximal and Distal expression values (Fig 2B). This difference was statistically significant and suggests that the Proximal and Distal enhancers are weakly synergistic with one another to drive higher levels of expression.

While enhancers are commonly conceptualized as Boolean logic gates that switch expression from OFF to ON based on the binary presence/absence of upstream factors, on a quantitative level this is thought to reflect fundamentally sigmoid transcriptional responses [40]. The functioning of gene regulatory networks is thus likely to be critically dependent on the quantitative details of these sigmoid transitions. As expected, we found that the Proximal, Distal and Full Length constructs all show roughly sigmoid responses to graded MAPK pathway inhibition (Fig 2C–2G). All three dose-response curves were, however, quite different from one another.

The Proximal and Distal enhancers both exhibited a simple monophasic relationship to MAPK pathway inhibition (Fig 2C), though with apparent differences in sensitivity and cooperativity. To quantify these differences, we fitted Hill functions (4-parameter logistic curves) to the dose-response data. As the expression data is skewed and heteroskedastic, we bootstrapped the nonlinear regression residuals to generate median parameter estimates and confidence intervals (S1 Fig). Two key parameters of these models are the $EC_{50}$ and the Hill Coefficient. The $EC_{50}$ is the drug concentration giving a half-maximal response, and is a direct measure of sensitivity. The Hill Coefficient is an exponential term describing the steepness of the transition between upper and lower plateaus, and is an implicit measure of cooperativity in which higher absolute values indicate greater cooperativity.

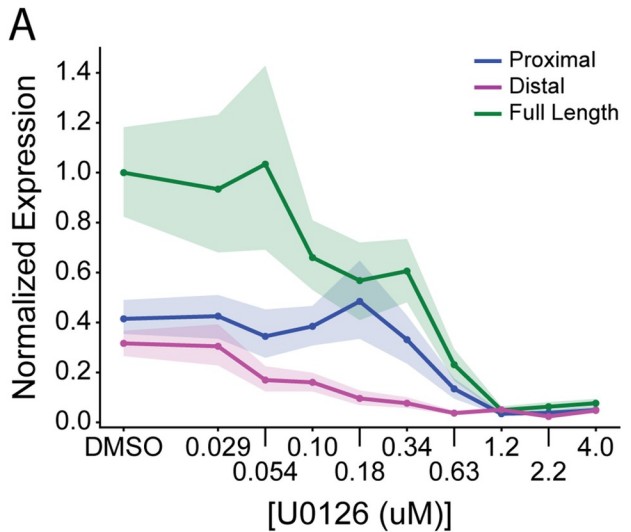

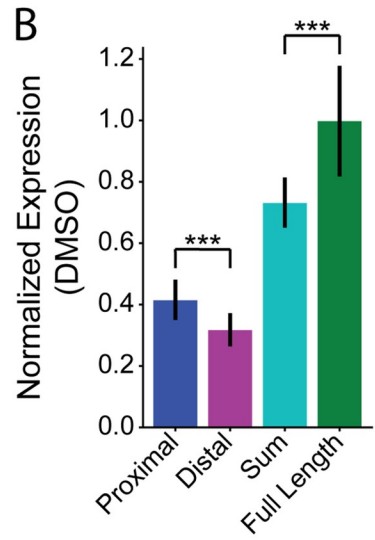

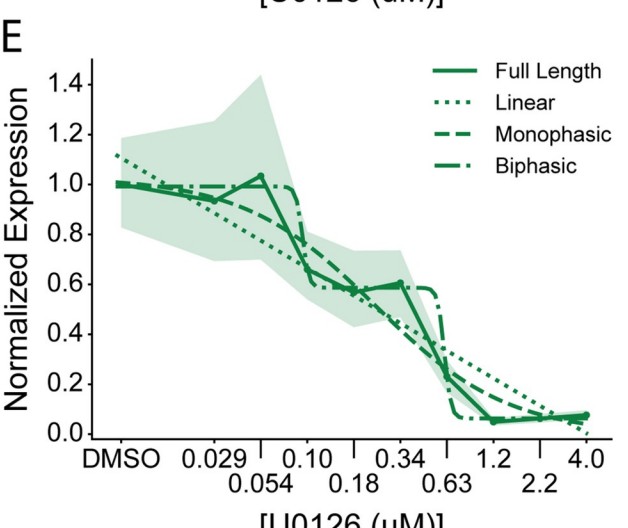

**D**

| Reporter | Parameter | Median | 95% CI | Mean |
|---|---|---|---|---|
| Proximal | Hill Coefficient*** | -5.00 | (-36.6,-3.34) | -9.49 |
| | EC50*** | 0.487 | (0.338,0.616) | 0.474 |
| Distal | Hill Coefficient*** | -1.43 | (-2.55,-0.807) | -2.07 |
| | EC50*** | 0.0686 | (0.0327,0.109) | 0.0659 |

**F**

| Curve Type | AIC | Relative Likelihood |
|---|---|---|
| Linear | -394.8 | 0.0832 |
| Monophasic | -397.7 | 0.349 |
| Biphasic | -399.8 | 1.00 |

**G**

| Parameter | Median | 95% CI | Mean |
|---|---|---|---|
| Hill Coefficient A | -26.9 | (-35.9,-1.03) | -22.2 |
| EC50 A | 0.0933 | (0.00851,0.102) | 0.0724 |
| Hill Coefficient B | -25.7 | (-37.6,-3.88) | -23.0 |
| EC50 B | 0.614 | (0.471,0.631) | 0.598 |

**Fig 2. Different *Bra* enhancers have distinct U0126 dose-response curves.** (A) Normalized expression values for each reporter plotted against U0126 dose. Shading indicates 95% confidence intervals of the mean. (B) Bootstrapped mean normalized expression of each reporter at the DMSO control dose, as well as the sum of the Proximal and Distal reporter. Error bars indicate 95% bootstrap confidence intervals (independent samples t-test; ***: p<0.0001). (C) Normalized expression values of the Proximal and Distal reporters, with best fit curves plotted using the median parameter values from bootstrapped curve fitting. Black point and error bars indicate the median $EC_{50}$ and its 95% bootstrap confidence interval for each construct. (D) Summary of Hill Coefficients and $EC_{50}$ parameter estimates for the curves in (C). Differences in parameters for the Proximal and Distal reporters were compared by Wilcoxon Rank-Sum test (***: p<0.0001). (E) Full Length reporter normalized expression values with linear, monophasic, and biphasic models fit to the data by nonlinear regression. (F) AIC and relative likelihood values for each model shown in panel (E). (G) Summary of Hill Coefficients and $EC_{50}$ parameter estimates of the A and B phases of the biphasic curve fit for the Full Length Reporter. $EC_{50}$ values in (D) and (G) were calculated in log space, but are shown here after conversion back to a linear scale.

As predicted from our qualitative assessment of the reporter imaging (Fig 1D), the $EC_{50}$ values of the Proximal and Distal reporters were very different from one another (Fig 2D). The Distal reporter's $EC_{50}$ of 0.0686 μM was less than 1/7th of the $EC_{50}$ of the Proximal reporter, and these differences had strong statistical support. The Hill Coefficients were also very different (Fig 2D), with a median parameter estimate of -5.00 for Proximal vs. -1.43 for Distal. There was again strong statistical support for these parameters being different. Together, these differences in expression strength, sensitivity and cooperativity all indicate that the Proximal and Distal enhancers are fundamentally different from one another in their quantitative responses to MAPK pathway activity despite their dependence on the same Zic and Ets input factors.

The Full Length construct showed a more complex U0126 dose-response curve that suggested a potentially biphasic relationship. To investigate this, we fitted both a monophasic Hill function as well as a biphasic double Hill function to the data. We also fitted a simple linear relationship (Fig 2E). We used the Akaike Information Criterion (AIC) to estimate the relative likelihood of the three different models. The biphasic model was preferred, but the relative likelihood differences were modest (Fig 2F). Parameter estimates for the Hill Coefficients of the two phases of the biphasic fit were quite high but had very broad confidence intervals (Figs 2G and S1B and S1C). A more finely graded series of U0126 doses would be needed to gain confidence as to whether these putative individual transitions are more or less cooperative than the single enhancer constructs. The $EC_{50}$ values of the two phases were quite close, however, to the $EC_{50}$ values of the Proximal and Distal constructs, supporting the idea that the two enhancers act in a quasi-additive fashion. A complete table of estimates of all bootstrap curve fit parameters can be found in S1 Table.

## Binary and graded responses at the single-cell level

The dose-response curves in Fig 2 are based on summed expression across the entire embryo, which is straightforward to automate and robust to the loss of embryonic morphology at higher doses of U0126 in which notochord cells cannot be reliably identified. Embryo-level measurements, however, may potentially obscure important differences in transcriptional responses at the level of individual cells. One possibility is that the transcriptional responses to MAPK inhibition might be more switch-like and quasi-Boolean at the single-cell level, and that the graded decreases in whole-embryo reporter expression seen at intermediate doses of U0126 might be largely a function of the fraction of cells in 'ON' versus 'OFF ' states. Alternatively, there could be a uniform but graded loss of expression across all the notochord cells (Fig 3A). These 'switch' and 'fade' mechanisms are not mutually exclusive and might represent two extremes on a spectrum of possibilities that ultimately reflect the cooperativity of transcriptional responses at the single cell level. Our major concern here was to exclude the possibility that the graded responses to MAPK inhibition observed when quantifying expression at the whole-embryo level were solely a function of the fraction of cells exhibiting a Boolean loss of all expression.

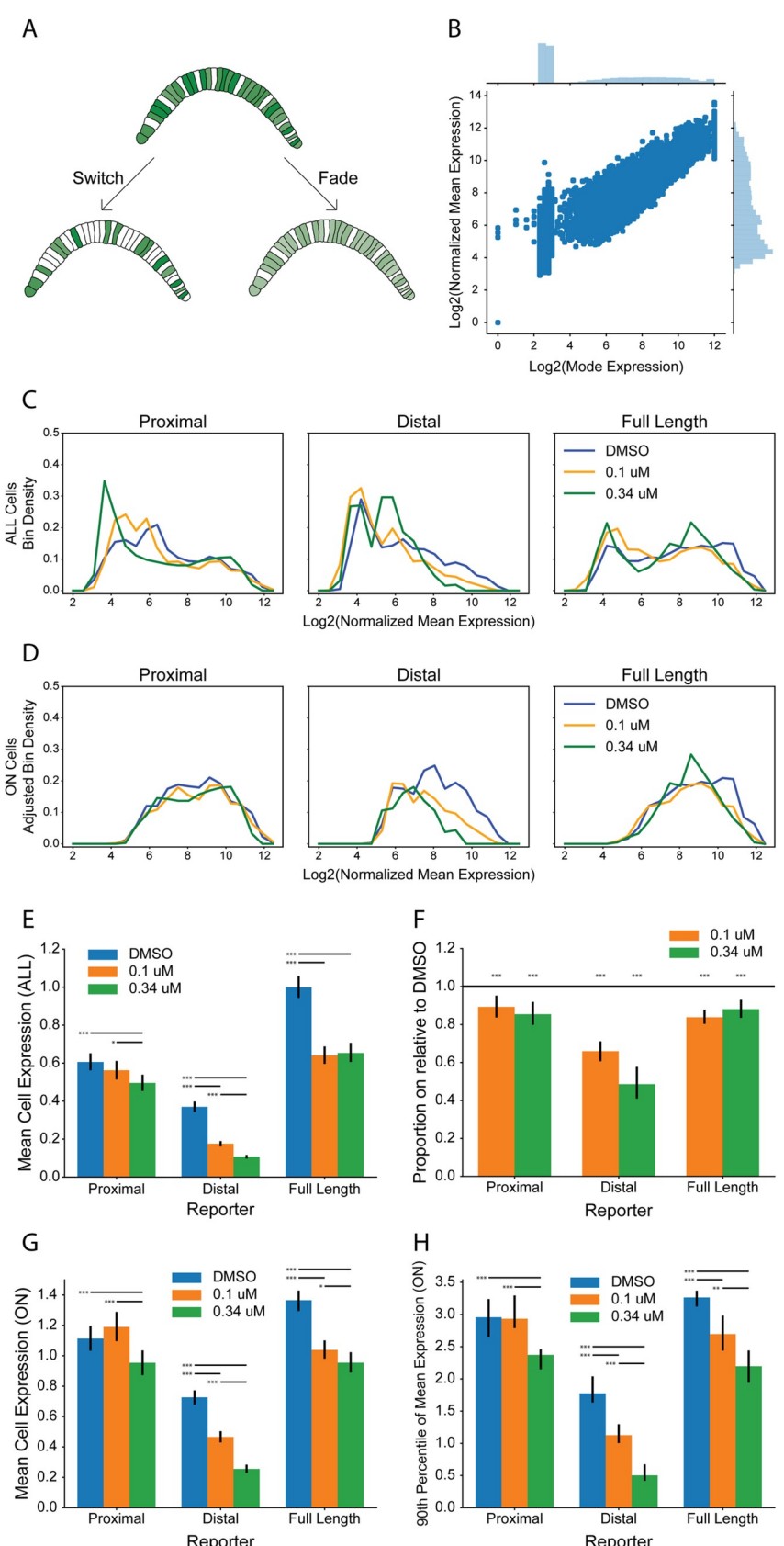

**Fig 3. Quantifying responses to MAPK inhibition at the single-cell level. (A)** Cartoon diagram of switch and fade models of single-cell expression loss at $EC_{50}$. Electroporated transgenes in *Ciona* are expressed mosaically, with no expression in some cells and variable expression in others. In the switch model, expression loss at $EC_{50}$ involves a complete loss of expression in half of the expressing cells. In the fade model, expression loss at $EC_{50}$ involves a graded ~50% weakening of expression in all the expressing cells. The switch and fade models represent two extremes of a continuum of possible mechanisms. **(B)** Scatterplot of mode pixel values versus normalized mean pixel values for disk-shaped regions of interest spanning each notochord cell nucleus. **(C)** Normalized mean cell expression distributions for all cells, regardless of whether they were classed as ON or OFF. **(D)** Normalized mean cell expression distributions for ON cells, scaled according to the proportion of cells that are ON. **(E)** Bar plot of mean normalized expression of ALL cells for each reporter at DMSO, 0.1 μM U0126, and 0.34 μM U0126. Error bars indicate 95% bootstrap confidence intervals; horizontal bars indicate significantly different pairwise comparisons of the bootstrap distributions, $^*p < .05$; $^{**}p < .005$; $^{***}p < 0.001$. **(F)** Bootstrap estimates of the decrease in the fraction of cells classed as ON at a given dose compared to matched DMSO controls. Error bars are 95% confidence intervals. The horizontal reference line is at a ratio of 1, which would indicate no change in the proportion of expressing cells. **(G)** Bootstrap estimates of changes in the mean of the distribution of single-cell normalized mean expression values. **(H)** Bootstrap estimates of changes in the 90th percentile of the distribution of single-cell normalized mean expression values. Error bars in **(G)** and **(H)** are 95% bootstrap confidence intervals of the median bootstrap value; horizontal bars indicate significantly different pairwise comparisons of the bootstrap distributions, $^*p < 0.05$; $^{**}p < 0.005$; $^{***}p < 0.001$.

To address this, we measured reporter expression in over 16,000 individual notochord cells from our confocal dose-response dataset at the DMSO, 0.1 μM, and 0.34 μM doses. The *Ciona* notochord consists of exactly 40 cells, so this represents ~400 embryos analyzed in total between the three reporters, three doses and multiple biological replicates. These doses were selected to be as close as possible to $EC_{50}$ for the different reporter constructs while still being able to reliably identify all notochord cells for quantitation. Mean whole-embryo expression of the Distal reporter at 0.1 μM dose represents 51% of its expression in the DMSO controls. The Full Length construct is expressed at 66% of control expression at 0.1 μM U0126 and 61% at 0.34 μM. These doses were also at approximately the start and end of the putative intermediate plateau for this construct. The $EC_{50}$ for the Proximal construct is beyond the dose at which notochord cells become unrecognizable, so the mean expression at 0.34 μM is only modestly decreased to 80% of control expression.

For each embryo, we manually identified the midpoint of each notochord cell nucleus in the Z dimension and then positioned a circular region of interest (ROI) of fixed area on top of it. The nucleus can be easily identified using the phalloidin channel based on perinuclear actin blobs and a lack of staining inside the nucleus compared to the faint signal in the cytoplasm. We used FIJI to measure various metrics of signal intensity from each nucleus midplane ROI, including the mean, median and modal grey values. Notochord cells were only measured if all notochord cells from the same embryo could be identified.

Given the mosaicism of electroporated transgene expression in *Ciona*, even DMSO control embryos typically have a considerable fraction of non-expressing cells and extensive variation in the brightness of expressing cells. We found that there was sufficient variation in both background intensity and the intensity of expressing cells that the mean gray value of each circular ROI did not clearly distinguish between ON and OFF states (Fig 3B). The distribution of modal gray values of these ROIs, however, was distinctly bimodal and manual inspection of cells in the two categories confirmed that this metric cleanly separated high background non-expressing cells from weak but *bona fide* expressing cells. We cannot exclude there being extremely faint reporter expression in some cells that cannot be separated from background staining, but any such staining would have to be very weak given the sensitivity and broad dynamic range of this assay.

The observed distributions of reporter intensity values are shown in Fig 3C (all cells) and Fig 3D (only ON cells). The distributions in Fig 3D have also been scaled to the proportion of cells that are ON relative to the DMSO dose, as shown in Fig 3F. Differences between U0126

doses are apparent, but these distributions are relatively noisy and complex. To better understand the phenomena driving expression loss at the single-cell level, we first quantified the overall response of the three reporter constructs in terms of the mean expression of all measured cells to confirm that this cell-based metric was comparable to the automated whole-embryo reporter quantitation (Fig 3E). One potential concern was that the whole-embryo quantitation includes ectopic expression in the mesenchyme cells or elsewhere whereas the cell-based measurements were specific to notochord cells. As expected, the Proximal reporter showed the least change over the doses measured, with the 0.34 μM dose having a small but significant loss of expression compared to DMSO or 0.1 μM, but no significant difference between DMSO and 0.1 μM. The Distal reporter had highly significant losses of expression at each increase in U0126 dose. The Full Length reporter had a large, highly significant loss in mean cell expression between the DMSO and 0.1 μM doses, but no significant difference between 0.1 and 0.34 μM doses. These findings were largely consistent with the whole-embryo data (Fig 2A), indicating that the embryo-level data are not biased by ectopic expression outside the notochord.

We next tested the possibility that expression loss at intermediate doses might involve a complete loss of expression in at least some cells by quantifying the fraction of cells that were inferred to be ON vs OFF based on the bimodal distribution of modal gray values. Analyses of the ON fraction were quite sensitive to experiment to experiment variation in electroporation efficiency, so we calculated bootstrap confidence intervals based on changes in the ON fraction between matched DMSO controls and the 0.1 μM and 0.34 μM doses (Fig 3F). We found that both these doses caused a statistically significant ($p<0.001$) decrease in the proportion of cells detectably expressing all three reporter constructs. This effect was weaker in magnitude for the Proximal and Full Length reporters, which showed only a 10–20% decrease in the ON fraction in U0126-treated cells compared to DMSO controls. This effect was stronger in magnitude for the Distal reporter, which showed a ~40% decrease in the ON fraction at 0.1 μM U0126, and a ~55% decrease at 0.34 μM. This demonstrates that at least part of the graded decrease in bulk expression seen at intermediate U0126 doses is due to an apparently complete loss of expression in a subset of notochord cells. We cannot exclude there being extremely faint expression in some of these cells that is below our threshold of detection, but this antibody-stained reporter imaging provides a sensitive readout.

We then tested the possibility that expression loss at intermediate doses might also involve a graded loss of expression in individual ON cells by quantifying both the mean and the 90th percentile of mean cell expression values of just the cells inferred to be ON (Fig 3G and 3H). The mean provides a standard and intuitive measure of central tendency in the cell-by-cell distributions of reporter intensity values. The 90th percentile was chosen to provide a specific but outlier-robust measure of whether the strongly expressing cells become quantifiably weaker at intermediate doses. We again used bootstrap estimates of these parameter values and their associated confidence intervals. The mean cell expression values of just the ON cells showed very similar trends at increasing U0126 doses to the changes in expression overall (Fig 3G). This indicates that the graded decreases in bulk expression seen at intermediate U0126 doses are at least in part a function of graded decreases in expression levels at the single-cell level and not merely a function of changes in the fraction of cells expressing the reporter at all. The 90th percentile values also showed similar trends, indicating that intermediate doses of U0126 cause graded decreases in the expression of even the brightest expressing cells (Fig 3H). It was also notable that the mean and 90th percentile cell expression of the Proximal reporter ON cells was close to or even higher than the mean or 90th percentile cell expression of the Full Length ON cells at all measured doses. Together, it is clear from these analyses that the loss of reporter expression at increasing doses of U0126 involves both an increase in the fraction of

cells that fail to detectably express the reporter and also a graded decrease in the intensity of expressing cells. This indicates that *Bra* reporter expression is not intrinsically Boolean over the time-scale examined. Graded responses to intermediate levels of MAPK pathway inhibition are seen at the level of single notochord cells.

## Endogenous *Bra* expression follows similar responses to U0126

While our reporter experiments revealed characteristic dose-response behaviors for each enhancer, reporter assays may be confounded by the lack of genomic context or normal chromatin structure. To test whether endogenous *Bra* expression has a similar U0126 dose-response behavior to the Full Length reporter construct, we treated unelectroporated embryos with a range of U0126 doses, fixing the embryos at the mid-gastrula stage (Hotta stage 12), and staining for *Bra* mRNA by *in situ* hybridization (Fig 4A–4H). Fixing the embryos at mid-gastrula allows for each notochord precursor blastomere to be accurately identified, thus allowing us to score each cell for *Bra* expression on a semi-quantitative integer scale, ranging from 0 (no expression) through 3 (robust expression).

When the average scores from each embryo were rescaled to the same DMSO average of 1 as the Full Length quantitative reporter data and plotted together, the two dose-response curves were quite similar (Fig 4I). The *in situ* curve was not as distinctly biphasic as the Full Length reporter curve, but the 95% confidence intervals were largely overlapping and the *in situ* data showed the same trend of a modest decrease over lower doses followed by a precipitous decline between 0.5 μm and 1 μm. These semi-quantitative *in situ* dose-response curves can only be superficially compared to the reporter dose-response curves, but this suggests that transient reporter assays are indeed a reasonable proxy for the effects of U0126 treatment on endogenous *Bra* expression. It remains possible, however, that there are other regulatory elements outside the Full Length region that contribute to endogenous *Bra* expression and that endogenous Bra regulation may involve aspects of chromatin structure that are not recapitulated using transient reporters.

Grouping the *in situ* scores into medial primary, lateral primary, and secondary notochord cell groups revealed small but clear differences in their responses to U0126, particularly at moderate doses between 0.25 μM and 1.0 μM (Fig 4J–4L). The medial A8.5 and A8.6 cell pairs have significantly higher expression at 0.5 μM U0126 than the more lateral primary notochord precursors A8.13 and A8.14 and the secondary notochord precursor B8.6. (Fig 4K and 4L). This indicates that the medial primary notochord cells are less sensitive to MAPK pathway inhibition than the more lateral notochord cells. A heatmap visualization of the *Bra in situ* dose-response data also demonstrates this effect and confirms that the loss of expression at intermediate doses of U0126 involves both graded decreases in expression in some cells as well as an apparently complete loss in others (Fig 4M–4R). This lateral-to-medial loss of expression is particularly evident when these heatmaps are compared to heatmaps in which the positions of the blastomeres have been randomly shuffled within each embryo (Fig 4M'–4R').

Having identified this unexpected differential sensitivity to MAPK signaling between different notochord founder cells, one question is whether there are differences in the behavior of the Proximal, Distal and Full Length reporter constructs in the notochord sublineages derived from these different founder blastomeres. All three constructs drive expression throughout the notochord, but it is possible that there are quantitative differences in the strength of expression or the response to U0126 treatment. This cannot be addressed using our whole-embryo dose-response data, because those measurements involved summing expression across entire embryos and not individual cells. It can be partially addressed, however, using our cell-by-cell measurements at the DMSO, 0.1μM and 0.342μM doses. These dose-response experiments

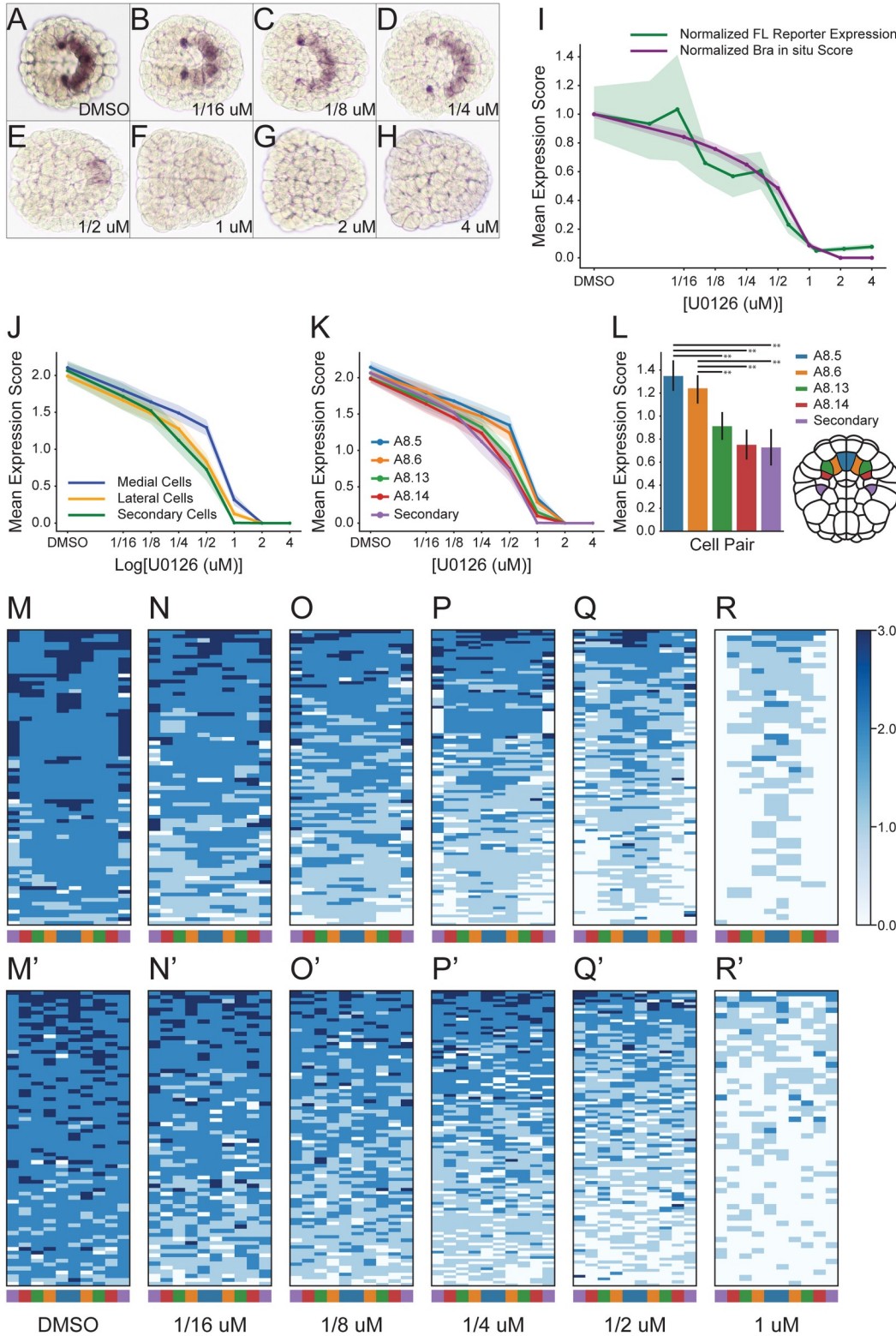

**Fig 4. Endogenous *Bra* expression in response to graded MAPK inhibition. (A-H)** Representative images of mid-gastrula stage embryos at the indicated doses stained by *in situ* hybridization for endogenous *Bra* expression. **(I)** U0126 dose-response curves for semiquantitative scores of endogenous *Bra* expression (purple) overlaid on the Full Length reporter whole-embryo expression data (green). Both dose-response curves are scaled to 1 at the DMSO control dose. Shading

indicates 95% confidence intervals. **(J-L)** Dose response curves for the semi-quantitatively scored endogenous *Bra* expression separated by founder cell identity. The scoring system involved subjectively classing individual cells on an integer scale between 0 (no expression) and 3 (very strong expression). Shading and error bars indicate 95% confidence intervals. **(J)** Cell scores for blastomeres A8.5 and A8.6 were plotted as medial, while scores for blastomeres A8.13 and A8.14 were plotted as lateral. **(K)** Plotting of individual blastomeres. **(L)** Clustered bar plots of the 0.5 μM dose. Horizontal bars indicate pairwise comparisons found to be significantly different by ANOVA followed by Tukey's HSD (**, p<0.005). Cartoon indicates location of cell pairs in the 110-cell embryo, anterior at top. **(M-R)** Heatmaps of embryos with nonuniform expression scores across the ten blastomeres, ordered from highest average score at top to lowest average score at bottom for **(M)** DMSO (n = 75), **(N)** 0.0625 μM (n = 72), **(O)** 0.125 μM (n = 87), **(P)** 0.25 μM (n = 109), **(Q)** 0.5 μM (n = 104), and **(R)** 1.0 μM (n = 54). Left to right ordering matches cartoon in **(L)**. Each row in these heat maps represents a different embryo and each column a different notochord founder cell. Embryos with uniform expression or no expression across all 10 cells were excluded. **(M'-R')** Similar heatmaps to **(M-R)** except that left-right cell order was randomized for each embryo.

were carried out at a stage when notochord intercalation was not yet complete. The final anterior-posterior position of each cell is therefore not unambiguous, but an approximate order could be inferred. In DMSO control treated embryos, there are clear differences between the three constructs as a function of anterior-posterior position (S2A Fig). The Full Length construct is distinctly stronger in the anterior and posterior notochord tips, with strongest expression in the posterior tip. The Proximal construct is stronger in the posterior tip. The Distal construct is more uniformly expressed but slightly graded towards the anterior. The anterior 32 cells in the intercalated notochord are derived from the primary notochord founder blastomeres, whereas the posterior 8 cells are derived from the secondary founder cells [41,42]. There is considerable stochasticity in the intercalation of the primary notochord cells, but the descendants of A8.5 and A8.6 are biased towards the anterior and the descendants of A8.13 and A8.14 are biased towards the posterior of the primary notochord [42,43]. It is likely that these AP differences in reporter construct expression reflect subtle expression differences between different notochord founder cells. These observations could potentially be confounded, however, by differences in notochord cell volumes, which become smaller towards the tips of the notochord, particularly in the secondary lineage at the posterior tip [43,44]. Our cell-by-cell measurements were made by quantifying a disk-shaped Region Of Interest within each cell and it is possible that reporter expression might be more concentrated in smaller cells without being different in total amount. Further studies in cleavage-arrested embryos could potentially resolve precise expression differences between different founder cells. We did not observe a strong difference in *Bra* reporter expression between primary and secondary cells in [45], but that study involved a later developmental stage and used a different quantitative approach. A stably integrated *Bra* reporter transgene in *Ciona savignyi* does show distinctly brighter expression in secondary cells during a comparable stage of mid to late intercalation [46].

We also quantified the reporter responses to U0126 in the cells where primary versus secondary lineage identity was relatively unambiguous (S2B and S2C Fig). All three enhancer constructs showed similar changes in response to 0.1 μM or 0.342 μM U0126 treatment when comparing primary and secondary notochord cells. A full U0126 dose-response series in cleavage-arrested embryos would be needed to rigorously identify differences in U0126 dose-response curves between different notochord sublineages, but there do not appear to be major differences between primary and secondary notochord for the constructs and doses tested here.

## Discussion

Some distributed enhancers are thought to respond to different combinations of direct upstream regulators [10]. Distributed enhancers that are dependent on different upstream

activators could potentially increase the robustness of development to many types of genetic, stochastic and environmental perturbation, and could also help to shape more complex transcriptional input/output relationships. An alternate but not mutually exclusive possibility is that distributed enhancers may shape transcriptional input/output relationships by having quantitatively different responses to the same directly upstream transcription factors. Our results here strongly support this hypothesis, as we find that the *Bra* Proximal and Distal enhancers have fundamentally different quantitative responses in a transient reporter assay to the graded inhibition of a MAPK-dependent signal acting directly upstream. The Proximal enhancer is less sensitive to MAPK inhibition but shows a sharper, more cooperative response, whereas the Distal enhancer is more sensitive but shows a more graded response. There may be other yet to be characterized upstream inputs that act on one of these enhancers and not the other, but their responses to graded inhibition of MAPK signaling are clearly and quantifiably different.

This study is subject to certain caveats. It is based on a transient reporter assay and not deletions of individual enhancers in the context of normal chromatin. It uses a protein reporter and not the MS2 RNA tagging system, which has not yet been implemented in *Ciona*. Pharmacological inhibition of MAPK-dependent FGF signaling will interfere not just with the direct induction of *Bra* by Ets family TFs but also any relevant FGF-dependent feedback or feedforward loops. These concerns apply mostly to questions of the additivity of the *Bra* Proximal and Distal enhancers. Although they appear to be slightly super-additive in our experiments, this was not our major focus and a more elaborate set of controls akin to [12] would be needed to fully explore this question. We have not, for example, excluded the possibility that the region between the Proximal and Distal enhancer elements might contain relevant transcription factor binding sites. These enhancers may also have inputs other than Zic and Ets family members. Our major conclusion, however, that the *Bra* Proximal and Distal enhancers have different dose-response relationships to MAPK pathway inhibition is largely robust to these concerns. It is clear that they act very differently from one another in this assay, regardless of whether the assay perfectly recapitulates all of the properties of the endogenous genetic elements.

Given that the *Bra* Proximal and Distal enhancers have quantifiably different responses to MAPK pathway inhibition, this raises intriguing questions about the functions of these different elements. One possibility is that they might be differentially involved in the initiation versus the maintenance of *Bra* expression. Different FGF ligands have been shown to have separable roles in *Bra* induction and maintenance [27], so this is certainly plausible. This could potentially be tested by dose-response reporter assays in which U0126 was applied and embryos were fixed across different stages, or by CRISPR or morpholino disruption of individual FGF ligands. Another possibility is that the two enhancers might have subtly different roles in different notochord lineages, though both enhancers are able to drive expression in both primary and secondary notochord. Our results here show that there are quantitative differences in expression as a function of AP position for the three reporter constructs that likely reflect expression differences between lineages, but more elaborate experiments in cleavage-arrested embryos would be needed to identify potential lineage-specific differences in $EC_{50}$ or Hill coefficient. It also remains to be determined whether the *Brachyury* Proximal and Distal elements are functionally redundant in terms of endogenous genetic deletions.

Another question is why, on a mechanistic level, these two different enhancers have such distinct dose-response relationships to MAPK pathway inhibition. Small differences in the number, affinity, order and spacing of transcription factor binding motifs have been shown to have major effects on the strength and tissue-specificity of expression in *Ciona* [47–51], including various mutations of the *Bra* Distal enhancer [28]. Differences in the number,

affinity, order and spacing of Ets sites and/or binding motifs for other transcription factors acting cooperatively with Ets family TFs presumably account for the quantitative differences between the two *Bra* enhancers. This could be tested experimentally but the combinatorial space of relevant mutations and treatments is quite large and it would benefit from a more scalable reporter assay than the *in toto* confocal imaging used here.

In addition to quantifying differences in the sensitivity and cooperativity of the *Bra* Proximal and Distal reporter constructs in response to MAPK pathway inhibition, we also explored the U0126 dose-response curve of endogenous *Bra* expression. This supported our conclusion from the reporter assays that the two enhancers acting together have a complex dose-response curve that is not well represented by a simple monophasic Hill function. Unexpectedly, this identified a differential sensitivity to U0126 between different notochord founder cells, with the medial primary notochord cells being particularly resistant to MAPK pathway inhibition. This could potentially involve differences in cell-cell contact surface areas between notochord founder cells and their FGF-expressing endodermal neighbors. Several cell fate decisions in the early *Ciona* embryo are thought to involve quantitative differences in cell contact surface between adjacent cells [52], so it may be that transcriptional input/output relationships are quite finely tuned even in the compact and stereotyped *Ciona* embryo where inductive events are thought to largely involve direct cell contacts and not long-range gradients.

Given the ubiquity of shadow enhancers across different animal species that often have quite different mechanisms for cell fate specification, it is important to understand how shadow enhancers contribute to *cis*-regulatory logic in multiple contexts. It is possible, for example, that fundamental aspects of enhancer cooperativity may be different between cell fate decisions involving direct cell contacts versus the unusual long-range gradients of maternal TFs seen in insects with syncytial early development or other long-range gradients of signaling molecules commonly seen in vertebrate development. The simple but stereotypically chordate *Ciona* embryo provides a new model for quantitative studies of *cis*-regulatory input/output relationships.

## Methods

### *Ciona* husbandry and embryology

Adult *Ciona robusta* (formerly *C. intestinalis* type A) were collected from San Diego harbor and shipped to KSU by Marine Research and Educational Products, Inc. (M-REP, San Diego, CA), before being kept in a recirculating tank filled with artificial seawater (ASW) until use. Fertilized embryos were obtained by sacrificing 3 adult animals for their eggs and sperm, which were then mixed for *in vitro* fertilization, and immediately dechorionated using standard procedures [53]. Dechorionated embryos were grown in ASW treated with 0.1% Bovine Serum Albumin (ASW+BSA) to minimize clumping. Embryos were incubated at 19.5 degrees Celsius and staged according to [17].

### Enhancer identification and cloning

Publicly available ATACseq data from *Ciona robusta* embryos [5] viewed through the genome browser on the ANISEED database [54] were used to identify the boundaries of the enhancers used in this study. Enhancer regions were PCR-amplified from a 2.2 kb enhancer/promoter region of plasmid *Bra*>Rfa-Venus (Newman-Smith et al., 2015). Primer sequences (underline indicates genomic sequence): Distal Forward (ACGTCTCGAGTCATTGAGGTTTTGTCGCCC), Distal Reverse (ACGTAAGCTTCTCCCCTTTTTAGTTTGATTGATG), Proximal Forward (ACGTCTCGAGTCACAATACAAACAAAATATTTTGAC), Proximal Reverse (ACGTAAGCTTTATAGGTTTGTAACTCGCACTGAG), Third Forward (ACGTCTCGAGTGCTAGACCGCCATCGC),

and Third Reverse (ACGTAAGCTT<u>CCTAATGACGTCACGAAACG</u>). The Full Length reporter was generated by PCR amplification using the Distal Forward and Proximal Reverse primers. All PCR-amplified enhancer regions were cloned into the XhoI/HindIII sites of pX2+bpFOG >UNC76:Venus (Stolfi et al., 2015).

## Reporter assays

Fertilized, dechorionated embryos were electroporated with 45 μg *Bra*(Region of interest)-bpFOG>Venus in a total electroporation volume of 800 μL, then washed into ASW+BSA. For drug treatments, plates of 3 mL ASW+BSA were supplemented with 1:1000 dilution of U0126 stock solutions (Sigma Cat. #662005-1MG) dissolved in DMSO to produce the indicated concentrations of U0126. Embryos were added to drug-treated seawater in 100 μl volumes at the 16-cell stage and fixed at the early tailbud I stage (Hotta Stage 19) in 2% paraformaldehyde/ASW overnight at room temperature. It was not feasible to split electroporated embryos between more than 8 drug treatments given the confocal time needed to image matched sets of doses and reporter constructs. The first round of experiments treated embryos with the doses between 0.1 μM and 4 μM. When it became clear that lower doses were needed, a second round of experiments with U0126 doses between 0.029 and 0.1 μM were run for each reporter. Because it was impractical to image more than 16 reporter/drug combinations in one experiment, the 3 different reporters were tested in overlapping sets of 2. At least three independent electroporations were performed for each reporter/drug dose combination.

Fixed embryos were stained using a rabbit polyclonal anti-GFP primary antibody (Fisher Cat. #A-11122) and goat anti-rabbit-555 secondary antibody (Fisher Cat. #A-21429). Phalloidin-488 (Fisher Cat. #A-12379) was used to stain cortical actin in the embryos. Stained embryos were mounted to poly-L-lysine-coated coverslips and cleared in Murray's Clear.

Embryos were imaged on a Zeiss 880 Laser Scanning Confocal Microscope using a 40X 1.3NA oil immersion objective under constant scan speed and laser power settings. Gain settings for the 555 reporter channel were also held constant across all embryos. The imaging settings were carefully optimized to have only minimal saturation of the brightest expressing cells while still being able to detect very faint expression. Pixel sizes were set to 0.32 μm/pixel, and z slices were made at an interval of 1.5 μm. All images were collected in 12-bit mode. The embryos to be imaged were selected based on embryonic morphology using the phalloidin staining as visualized by widefield epifluorescence through the microscope eyepieces. The embryonic phenotypes induced by increasing doses of U0126 have characteristics that can generally be distinguished from embryos that have poor development due to dechorionation, electroporation or poor egg quality. Embryos representative of each dose were selected based strictly on embryonic morphology while excluding embryos with other malformations. The experimenter remained completely blind to reporter expression until after these representative embryos for 3D confocal imaging were chosen.

## Whole-embryo reporter quantification

Files containing the confocal stacks of embryos were passed to an in-house Python function that subtracted a fixed background level, applied a light median filtering to the reporter channel, and sum-projected the values of the phalloidin and reporter channels in the z axis. These flattened images sometimes contained false signal from specks of dust or other embryos intruding into the field of view. To ensure that we only quantified reporter signal from within single embryos, the script thresholded the phalloidin channel and used binary morphology to generate a binary mask approximating each embryo. All masks were individually checked, then edited by hand in FIJI/ImageJ [55] if they did not accurately capture the boundary of the

embryo. A summed enhancer value (SEV) was calculated for each embryo by summing all pixel values within each final mask region.

To account for variation of electroporation efficiency, the SEV scores of the DMSO control embryos for each electroporation were averaged, and the individual SEV score of each embryo in that same electroporation was divided by this average to obtain an experiment normalized score. This experiment normalized score was then multiplied by the average SEV of all DMSO control embryos for each reporter across all experiments to obtain a normalized scaled SEV score (NSSEV), and expressed as a fraction of the Full Length DMSO mean NSSEV. These normalized expression values were used in all plotting and curve-fitting calculations. Bootstrapping of DMSO expression scores was performed by resampling normalized expression values of each reporter 1000 times, ensuring that scores were sampled equally from each electroporation. Sum of Proximal and Distal reporters was determined by adding the Proximal and Distal average normalized expression values once for each bootstrap replicate to obtain a set of 1000 sums. Differences between the Proximal/Distal and Sum/Full Length pairs were tested by independent sample t-test. 95% bootstrap confidence intervals were calculated as the $2.5^{th}$ and $97.5^{th}$ percentiles for all of the bootstrapped parameter estimates.

Curves describing linear, monophasic, or biphasic relationships were fit according to the following formulas, where y is normalized expression, and x is the log10 of the U0126 concentration (DMSO was coded as two log-steps lower than the lowest U0126 dose to avoid a log of zero error):

Linear: $y = mx+b$

(m is the slope and b is the y-intercept)

Monophasic : $y = \frac{A-D}{1+10^{(C-x)*B}} + D$

(A and D are the top and bottom plateaus, respectively, B is the Hill Coefficient, and C is the $EC_{50}$.)

Biphasic : $y = \frac{A-D}{1+10^{(C-x)*B}} + \frac{D-G}{1+10^{((F-x)*E)}} + G$

(A, D, and G are the top, middle, and bottom plateaus, respectively; B and C are the Hill Coefficient and $EC_{50}$ of the first phase, respectively; and E and F are the Hill Coefficient and $EC_{50}$ of the second phase, respectively.)

Nonlinear regression fitting monophasic, biphasic, and linear curves was performed using the optimize.curve_fit function of the Scipy Python library [56]. We used minimally restrictive parameter bounds and crudely estimated parameter seeds to obtain a matrix of residuals from initial curve fits. These residuals were bootstrapped 1000 times to obtain bootstrap replicates, which were each used to fit a set of parameters by nonlinear regression. Differences in parameter values for Proximal and Distal reporter monophasic curves were determined by Wilcoxon Rank-Sum test. For Full Length reporter bootstrapping, bootstrap replicates that failed to be fit to a curve were dropped. Median values from each parameter distribution were used to plot the best fit curves for the Proximal and Distal reporters in Fig 2C.

## Individual cell measurement reporter quantification

Individual notochord cell measurements were made by opening confocal stacks in FIJI/ImageJ, identifying the z plane representing the approximate midpoint in Z of each notochord cell nucleus, and using the ROI Manager tool to manually place circular Regions of Interest (ROIs) with 10-pixel diameters over the nucleus of each notochord cell. The notochord cell nuclei are clearly distinguishable in the phalloidin channel. ROI measurements included reporter channel mean, standard deviation, mode, maximum, minimum, median, and total expression values for each cell, and were aggregated across all measured embryos for each given dose/reporter combination. Mean expression values were normalized and scaled in the same

manner as whole-embryo expression. We added a pseudocount of 1 prior to all log transformations of various expression metrics to avoid log of zero errors. Cells with a ROI mode pixel value of 10 or less were classed as being 'off' based on the bimodal distribution of the ROI modes and manual inspection of a subset of cells.

Bootstrapping of the ON/OFF ratios was performed by resampling cells from each U0126 treatment for each reporter 1000 times. Bootstrapping of matching DMSO-treated cells was performed by resampling matching numbers of DMSO cells from each experiment in the corresponding U0126 dose, generating a unique matched control set for each U0126/reporter combination for each bootstrap replicate. The ON ratio was calculated as the proportion of measured cells that were on in the U0126 bootstrap replicate divided by the same proportion in the matching DMSO replicate. Differences between these ratios and a ratio of one were tested by t-tests comparing each reporter/dose combination's bootstrap distribution to 1.

Bootstrapping of normalized mean expression values was performed by resampling cells at each dose 1000 times, ensuring that the number of cells resampled from each electroporation equaled the number of cells measured in the same electroporation, bootstrapping separately for the ALL cells and ON cells only groups. Statistical tests for differences between doses for the mean and 90[th] percentile values were performed by matching bootstrap replicates for each pairwise U0126 dose comparison within each reporter, counting the number of cases in which expression at the higher dose exceeded expression at the lower dose, and dividing by 1000 to obtain a p-value.

### *in situ* hybridization

Fertilized, dechorionated embryos were allowed to develop in ASW+BSA at 19.5 degrees Celsius until the 16-cell stage, at which point they were drug treated in the same manner as for the reporter assays. Embryos were fixed in MEM-PFA at mid-gastrula (Hotta stage 12) on ice 10 minutes before overnight storage at 4 degrees. Probe synthesis and *in situ* hybridization was conducted as in [57]. Mounted embryos were imaged on an Olympus BX61WI compound microscope using a 10X 0.3NA objective and a Canon EOS Rebel T3i digital camera under constant illumination conditions. Images were opened in FIJI/ImageJ and each notochord precursor cell was scored on a whole-number scale from 0 (no expression) to 3 (robust expression). Line plots and bar plots show mean scores with 95% confidence intervals. Differences in mean expression scores were calculated by one-way ANOVA followed by Tukey's HSD, looking within each dose for differences between cell pair scores.

Expression heatmaps were generated by ordering embryos from highest average expression to lowest average expression across all ten precursor cells. Embryos that had uniform scores across all ten precursor cells were not included in the heatmaps. Randomized heat maps were generated by shuffling the position of each precursor cell without replacement in each embryo.

### Data visualization and analysis

All statistical tests, plotting, curve fitting and visualization of quantitative expression data were performed in Python, using standard Pandas, Numpy, Matplotlib, Scipy, and Seaborn packages.

### Supporting information

**S1 Fig. Parameter distributions from bootstrapped residuals.** (A-C) Bootstrap parameter distributions for curves fit to the (A) Proximal, (B) Distal, and (C) Full Length whole-embryo reporter data. Monophasic Hill functions are fit for the Proximal and Distal constructs. A

biphasic double-sigmoid function is fit for the Full Length construct.
(TIF)

**S2 Fig. AP and Primary/Secondary differences in reporter expression.** A) Normalized reporter expression as a function of approximate anterior-posterior position in DMSO-treated control embryos. B) Normalized expression of the Proximal, Distal and Full Length reporter constructs at the indicated U0126 doses in primary notochord cells. C) Normalized expression of the Proximal, Distal and Full Length reporter constructs at the indicated U0126 doses in secondary notochord cells.
(TIF)

**S1 Table. Full bootstrap parameter estimates for reporter dose-response curve fits.**
(XLSX)

**S1 Appendix. Zip archive of data files and analysis scripts.**
(ZIP)

# Acknowledgments

The authors thank Konner Winkley for assistance with Python programming. We also thank the KSU CVM Confocal Microscopy Core facility for the use of their microscopes.

# Author Contributions

**Conceptualization:** Matthew J. Harder, Michael T. Veeman.

**Formal analysis:** Matthew J. Harder, Michael T. Veeman.

**Funding acquisition:** Michael T. Veeman.

**Investigation:** Matthew J. Harder, Julie Hix, Wendy M. Reeves, Michael T. Veeman.

**Project administration:** Wendy M. Reeves.

**Resources:** Wendy M. Reeves.

**Supervision:** Wendy M. Reeves, Michael T. Veeman.

**Writing – original draft:** Matthew J. Harder, Michael T. Veeman.

**Writing – review & editing:** Matthew J. Harder, Wendy M. Reeves, Michael T. Veeman.

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
