## [Decision Letter · Decision Letter 0]

20 Oct 2020

Dear Dr Veeman,

Thank you very much for submitting your Research Article entitled 'Ciona Brachyury proximal and distal enhancers have different FGF dose-response relationships' to PLOS Genetics. Your manuscript was fully evaluated at the editorial level and by independent peer reviewers. The reviewers appreciated the attention to an important problem, but raised some concerns about the current manuscript. Based on the reviews, we will not be able to accept this version of the manuscript, but we would be willing to review a revised version. We cannot, of course, promise publication at that time.

If you decide to revise the manuscript for further consideration at PLOS Genetics, please aim to resubmit within the next 60 days, unless it will take extra time to address the concerns of the reviewers, in which case we would appreciate an expected resubmission date by email to plosgenetics@plos.org.

[LINK]

We are sorry that we cannot be more positive about your manuscript at this stage. Please do not hesitate to contact us if you have any concerns or questions.

Yours sincerely,

Gregory P. Copenhaver

Editor-in-Chief

PLOS Genetics

Reviewer's Responses to Questions

**Comments to the Authors:**

Reviewer #1: The manuscript by Harder et al. 'Ciona Brachyury proximal and distal enhancers have different FGF dose-response relationships' is a very interesting study that convincingly supports the idea that shadow/distributed enhancers are not simply redundant elements. The authors have made use of simple methods (GFP as reporter in transcriptional assays and colorimetric in situ hybridization) to provide quantitative measurements of differential response to Fgf/Erk pathway. In addition, they have shown that endogeneous Brachyury expression has different sensitivities to MAPK inhibition depending on notochord precursors. It would thus be important to determine whether the proximal, distal and 'full length' enhancers also display such differential sensitivity.

Reviewer #2: Comments uploaded as an attachment

Reviewer #3: The authors of “Ciona Brachyury proximal and distal enhancers have different FGF dose-response relationships” demonstrate that two “distributed” Brachyury regulatory elements that drive the same expression pattern behave differently in response to variations in one of the inputs (FGF/MapK signaling). The difference in response, overall, is clear, and they also attempt to ascertain if there are non-Boolean dynamics in play regarding each elements’ response to MAPK interference. Furthermore, they demonstrate that native Brachyury expression is impacted similarly, and that some notochord lineages respond to MAPK inhibition differently than other notochord lineages. It is an interesting piece of work and with moderate revisions will help decipher FGF-induced transcriptional dynamics and more broadly make an important contribution towards understanding the role of “distributed” seemingly redundant enhancers in regulating gene expression.

This manuscript would be acceptable given the following issues are addressed:

1. Line 233 combined with line 243 reads as if over 16,000 individual notochord nuclei midpoints in the Z dimension were manually identified. Is that a correct interpretation? That could be made clearer and as it seems exceptionally high perhaps include reasoning as to why that many were needed (e.g. statistical robustness). The “cell measurements” excel sheet in the appendix provides a clue to what was counted (40cells x 10 embryos x # of treatments x each reporter construct) but it should be briefly mentioned.

2. The authors state that they “only included replicates where the capacitance reported by the electroporator in time constant mode was between 900 and 1300 μFd.” It would be helpful to include a description of why these parameters were used (how exceptions create unfavorable conditions/increased mosaicism if that is the case), as well as include a relevant reference.

3. Regarding the differential responses by the two promoters, what jumped out was the lack of description of ETS binding site quantities/distributions per element. Activated ETS proteins are the effective readout for FGF/MAPK signaling, and as the authors mention in the discussion the main difference in how the two promoters respond likely have to do with different quantities of ETS binding sites. It would be simple enough to generate a basic diagram of the binding sites and discuss responses versus ETS/ZicL binding site distribution.

Furthermore, hypothetically if the distal promoter had two ETS sites while the proximal had five ETS sites, hypotheses could be drawn and discussed regarding why MAPK inhibition induced a switch or fade response in the proximal or distal element.

4. The description of switch vs. fade is unclear. The authors performed multiple layers of statistical analysis on their data to attempt to show that different promoters exhibited different switch vs. fade responses. But the writing surrounding the topic (the description of the results and how they were obtained), is not clear enough as it is written currently. It is entirely possible that a simple rewrite of that section will clear the matter up. What this reviewer suggests might help are descriptions of why certain data was analyzed. For example, what is it about the mean and the 90th percentile data that make it most suitable for secondary analysis?

Furthermore, it should be explained to the reader how you can be sure an ON->OFF “switch” is not simply the result of a lower-expressing cell “fading” below the threshold set by the authors as ON or justify the decision use this method despite this issue. Because of the variability in TG expression patterns, it is important that the reader is guided through the information on which conclusions are drawn in order to be properly convinced.

5. The authors detail how image files from earlier experiments were analyzed for detection of switch vs fade mechanisms of response. However, the authors fail to mention how the actual nuclei were detected. If the fluorophores driven by each promoter were UNC76-tagged as they state, nuclear staining, if present at all, would not be more intense than the surrounding cytoplasm. This is evidenced by the images therein, as well as in all previously published reports of UNC76-tagged fluorophores. The authors must address this question.

6. The authors present interesting data regarding how during gastrulation the notochord founder cells have a variable response to MAPK inhibition via U0126. Effectively, it appears the sensitivity of Bra expression to MAPK inhibition decreases moving from medial cells to the lateral cells and secondaries. Notochord cell populations can be traced back to either primary or secondary lineages (as in Corbo et al, 2006). The authors should explain why they did not choose to analyze the 2° notochord lineage (posterior cells) as a distinct population from the 1° lineage. Without any new experiments the authors should be able to provide an analysis of reporter expression in response to U0126 in the 2° notochord lineage cells (especially as the authors demonstrate that Bra expression in 2° lineage is most sensitive to U0126). This could be a valuable addition and would be simple enough to perform.

7. Line 438 states that “Fertilized, dechorionated embryos were electroporated with 45μg…”. This is only relevant if the reader is provided a volume of eggs + seawater electroporated.

8. The authors state that it was “not feasible to split electroporated embryos between more than 8 drug treatments.” It is left to the imagination as to why, as many thousands of embryos can be generated from a single transgene electroporation. This could easily be cleared up. Due to too few embryos per electroporated batch? Due to space or time constraints?

9. Regarding the switch vs. fade section and paragraphs in the discussion, it might be helpful to include a diagram of responses, similar to Figure 3A. Figure 3A is instructive, but also misleading as it disguises the complexity of the actual situation. It would be helpful to first show a more realistic cartoon of “average” TG expression (some empty notochord cells, some with low or med or high expression, ordered essentially at random), followed by average depictions of the divergent outcomes, per TG tested. While this may not capture the completeness of the story it could be illustrative to a meaningful degree.

10. It is clear that the authors went to trouble to attempt to control for electroporation variation when comparing the three promoters response to concentrations of U0126. However, this reviewer wonders if there was a missed opportunity for the simple control of co-electroporating all three constructs if each TG expressed a different color. It has been demonstrated (Chen et al, 2012) that when electroporated in equimolar amounts transgenes produce consistent ratios of signal per embryo. Thus, if each Bra promoter drove a different color, the separate intensities could be compared in single embryos with only the U0126 treatment as a variable to assess, for intensity ratio of each color, per embryo, would be consistent in each electroporation.

**Have all data underlying the figures and results presented in the manuscript been provided?**

Reviewer #1: Yes

Reviewer #2: Yes

Reviewer #3: Yes

PLOS authors have the option to publish the peer review history of their article (what does this mean?). If published, this will include your full peer review and any attached files.

Reviewer #1: No

Reviewer #2: No

Reviewer #3: **Yes: **Bradley Davidson (I also think my post-doc C.J. Pickett should receive credit and he has agreed on this point)

---

## [Editor Report · Decision Letter 1]

10 Dec 2020

Dear Dr Veeman,

We are pleased to inform you that your manuscript entitled "Ciona Brachyury proximal and distal enhancers have different FGF dose-response relationships" has been editorially accepted for publication in PLOS Genetics. Congratulations!

Yours sincerely,

Gregory P. Copenhaver

Editor-in-Chief

PLOS Genetics

Comments from the reviewers (if applicable):

**Data Deposition**

http://datadryad.org/submit?journalID=pgenetics&manu=PGENETICS-D-20-01286R1

**Press Queries**

---

## [Editor Report · Acceptance letter]

13 Jan 2021

PGENETICS-D-20-01286R1 

* Ciona Brachyury* proximal and distal enhancers have different FGF dose-response relationships 

Dear Dr Veeman, 

We are pleased to inform you that your manuscript entitled "* Ciona Brachyury* proximal and distal enhancers have different FGF dose-response relationships" has been formally accepted for publication in PLOS Genetics! Your manuscript is now with our production department and you will be notified of the publication date in due course.

With kind regards,

Melanie Wincott

PLOS Genetics

On behalf of:
